# The Effect of Respiratory Muscle Training on the Pulmonary Function, Lung Ventilation, and Endurance Performance of Young Soccer Players

**DOI:** 10.3390/ijerph17010234

**Published:** 2019-12-28

**Authors:** Krzysztof Mackała, Monika Kurzaj, Paulina Okrzymowska, Jacek Stodółka, Milan Coh, Krystyna Rożek-Piechura

**Affiliations:** 1Department of Track and Field, University School of Physical Education, Wroclaw, Ul. Paderewskiego 35, 51-612 Wrocław, Poland; jacek.stodolka@awf.wroc.pl; 2Faculty of Physiotherapy, University School of Physical Education in Wroclaw, Poland, Ul. Paderewskiego 35, 51-612 Wrocław, Poland; monikakurzaj@wp.pl (M.K.); okrzymowska.paulina@gmail.com (P.O.); krystyna.rozek-piechura@awf.wroc.pl (K.R.-P.); 3Faculty of Sport, University of Ljubljana, Gortanova ul. 22, 1000 Ljubljana, Slovenia; Milan.Coh@fsp.uni-lj.si

**Keywords:** soccer, inspiratory training, lung ventilation, muscle strength, endurance, running

## Abstract

This study investigated whether the addition of eight weeks of inspiratory muscle training (IMT) to a regular preseason soccer training program, including incremental endurance training (IET), would change pulmonary function, lung ventilation, and aerobic performance in young soccer players. Sixteen club-level competitive junior soccer players (mean age 17.63 ± 0.48 years, height 182 ± 0.05 cm, body mass 68.88 ± 4.48 kg) participated in the study. Participants were randomly assigned into two groups: experimental (n = 8) and control (n = 8). Both groups performed regular preseason soccer training, including endurance workouts as IET. In addition to this training, the experimental group performed additional IMT for eigght weeks with a commercially available respiratory muscle trainer (Threshold IMT), with a total of 80 inhalations (twice per day, five days per week). Pre- and post-intervention tests of pulmonary function, maximal inspiratory pressure, and the Cooper test were implemented. Eight weeks of IMT had a positive impact on expiratory muscle strength (*p* = 0.001); however, there was no significant effect on respiratory function parameters. The results also indicate increased efficiency of the inspiratory muscles, contributing to an improvement in aerobic endurance, measured by VO₂max estimated from running distance in the cardiorespiratory Cooper test (*p* < 0.005).

## 1. Introduction

The effectiveness of soccer players depends on morphological, functional, and psychological factors, which determine appropriate playing tactics and success in soccer matches [1,2,3]. Consequently, soccer is widely recognised to be a prolonged, high-intensity, intermittent activity that requires players to perform regular, repeated sprints throughout the 90-minute game, and where the average exercise intensity is close to the anaerobic threshold of 80–90% of maximum heart rate (HR) [4,5,6,7,8]. The ability to perform intense exercise declines towards the end of a match, as well as immediately after the most intense periods of the game [9,10]. 

In this context, the implementation of a well-developed aerobic fitness training program helps soccer players to maintain repetitive high-intensity actions during a soccer match, accelerate their recovery process, and maintain their physical condition at an optimum level throughout the entire game and competition season [11]. One of the basic and most important parameters of motor preparation in soccer is cardiovascular fitness, which is often defined as aerobic endurance [12,13]. In most sports, including soccer, a type of workout is endurance running performance [14,15], often referred to as incremental endurance training (IET). It’s mainly based on aerobic efforts [14], but at the climax of its development, may fall into mixed aerobic-anaerobic energy release training, where the level of training intensity is 80–85% of maximum HR [16,17]. This type of training takes on special significance during the preseason and is the subject of the abovementioned research experiments.

Aerobic endurance performance is dependent on three important components: maximal oxygen uptake (VO₂max), anaerobic threshold, and work economy [18]. VO₂max is an important component, defined as the highest oxygen uptake that can be achieved during dynamic exercise with large muscle groups [19]. Therefore, VO₂max is an extensively used index for measuring the aerobic fitness of athletes and can be determined in both laboratory and field tests [16,20,21]. Some studies have indicated a significant relationship between VO₂max and distance covered during a match [21].

During exercise, respiratory muscles are subject to fatigue which limits their ability to work optimally, translating into insufficient oxygen supply to the working muscles [22]. Therefore, it seems appropriate to draw attention to the work of respiratory muscles in sports training, including the training of soccer players [23]. Inspiratory muscle training (IMT) applies an additional load to the diaphragm, being an accessory to the inspiratory muscles, to enhance their strength and endurance. Some research in the literature has assessed the effects of IMT in soccer players [24,25,26,27], demonstrating that the inclusion of such training in the primary soccer training process can improve inspiratory muscle strength (PI_max_), expiratory muscle strength (PE_max_), and exercise tolerance, and also reduce blood lactate (BLa) levels after a cycle of exercises [26]. Research conducted by Archiza et al. [28] on a group of 18 female soccer players indicated that IMT could relieve the metabolism of inspiratory muscles and, as a consequence, improve muscle oxygen supply during high-intensity exercise. This process can be translated into an improvement in fatigue tolerance and running efficiency of soccer players [28]. Consistent with this, Ozmen et al. [27] evaluated the effect of five weeks of IMT on the respiratory system function and aerobic endurance of soccer players, finding a significant improvement in respiratory muscle strength for the tested respiratory parameters. However, respiratory training did not significantly improve the soccer players’ tolerance to high-intensity exercise. The essence of this type of training is to cause an appropriate amount of resistance during inspiration, with simultaneous quiet exhalation. An important part of this training is the correct breathing technique [29]. Inspiratory muscle training can be an important procedure in sports training. It increases an athlete’s physical performance, reduces the concentration of blood lactate, and improves the ventilation of the lungs, diaphragm, and inspiratory muscles [30,31,32,33].

With the exception of PI_max_ and PE_max_, the peak expiratory flow (PEF), also called peak expiratory flow rate (PEFR), plays an important role in the assessment of pulmonary function. A measurement PEF is especially useful for assessing bulbar-innervated, inspiratory, and expiratory muscle function [34,35]. The PEF ranges from 500 to 700 litres/min for men and from 380 to 500 litres/min for women, and from 150 to 840 litres/min for children and adolescents, with variations due to age, race, and sex [36]. Peak flow readings are higher when a person is healthy, and lower when the airways are constricted. Lung functionality can be determined from changes in recorded values [34]. 

A problem to be considered when designing a well-balanced cardiovascular fitness training program for soccer players is that the maximal oxygen uptake is most effectively trained at an intensity of 90–95% of HRmax, which is generally achieved by running intervals. It is also believed that for VO_2_max to be improved, playing only soccer games is not enough as it does not provide sufficient exercise intensity over time [14,18,20]. It seems reasonable that this type of workout may represent excessive training stimuli for junior soccer players. Therefore, a combination of breathing training with a more intense running training, not exceeding 85% of the anaerobic threshold, should compensate for the intermittent activity discussed above (at 90–95%). Thus, this study aimed to determine whether the addition of eight weeks of IMT to a regular program of preseason soccer training, including IET, would change pulmonary function, lung ventilation, and aerobic performance in young soccer players. 

## 2. Materials and Methods

### 2.1. Participants

Sixteen club-level (Football Academy) competitive junior soccer players (mean age 17.63 ± 0.48 years, height 182 ± 0.05 cm, body mass 68.88 ± 4.48 kg) participated in this study. Participants were randomly assigned into two groups: inspiratory muscle training (IMT, n = 8) and control group (n = 8). Medical examinations showed no history of pulmonary disease in the participants and acknowledged their good health condition to participate in the study. All participants were non-smokers (self-report), with no evidence of respiratory restrictions or obstruction upon examination of the maximum flow-volume loops. The experimental protocol was approved by the University Institutional Ethics Committee, and all participants and their parents provided written informed consent for voluntary participation in the study. Before the study commenced, the participants were informed of the procedures and potential risks of the training.

### 2.2. Experimental Design and Task 

Participants were randomly assigned to the experimental or control group. During the 8-week experimental period, both groups continued their regular preseason soccer training program including endurance workouts as IET. Regular soccer training sessions were performed four times per week under the supervision of a coach. Supervised training sessions consisted of about 2 hours of soccer activities, including strength conditioning, drills for skill improvement, team scrimmaging, and sparring. The soccer training program was developed by the Football Academy. The experimental group performed additional IMT with a commercially available respiratory muscle trainer (Threshold IMT; Philips Respironics, Inc., Murrysville, PA, USA). The players were instructed not to participate in any physical activities during the study period other than those stated. The experiment required an assessment of respiratory muscle strength, pulmonary function, and indirect aerobic endurance (VO₂max) via an estimated performance of the Cooper test. The tests were carried out at similar times of the day (morning) for both groups, although the IMT test for the experimental group was performed first. The Cooper test was performed after that. Both tests were performed before the respiratory and running endurance intervention training was started. The initial performance assessment took place at the beginning of January when the participants were starting the preparation phase for their spring/summer competition season. Participation in the experiment required a familiarisation session with the test procedure. The participants were instructed to adhere to their usual diet and to not engage in strenuous activity the day before testing. On the test day, the participants were asked to not eat for at least 2 h before testing. For each participant, testing was scheduled at a similar time of day (±0.5 h) to minimise the effects of diurnal fluctuation.

### 2.3. Measures 

#### 2.3.1. Dynamic Lung Function

Pulmonary function (forced flow-volume loops) was assessed using a spirometer (Flowscreen; Jaeger, Wuerzburg, Germany) with a special adapter (780, 578, version 1.3). Measurements were made according to the recommendation of the European Respiratory Society [37]. During testing, the participants assumed their normal sitting position. The following variables were determined for all participants before and after IMT: vital capacity (VC), forced vital capacity (FVC), forced expiratory volume in first second (FEV_1_) and peak expiratory flow (PEF). 

#### 2.3.2. Maximal Inspiratory Pressure

A spirometer (Jaeger, Wuerzburg, Germany) with a shutter module was used for analysis of muscular respiratory pressure. The obtained values were expressed as a percentage of the normal values. The force of respiratory muscles can be evaluated using static measurements (PI_max_ and PE_max_) or dynamic manoeuvres (maximal voluntary ventilation, MVV). PI_max_ represents the highest sub-atmospheric pressure that can be generated during an inspiration against a blocked airway (Muller manoeuvre). PE_max_ is the highest pressure that can be achieved during a high expiratory effort against a blocked airway (Valsalva manoeuvre). These methods are usually performed by starting the manoeuvre from the residual volume (for MIP determination) or from the maximal capacity (for PE_max_ determination). There are a few contraindications for these exploratory manoeuvres: aneurysm, uncontrolled hypertension, urinary infection, and recent abdominal, or thoracic surgery. The participants underwent three to five maximal acceptable and reproducible manoeuvres (with differences of 3–9% between values). For the statistical evaluation, the maximal value obtained from these successive trials was taken into consideration. A 1-min interval was permitted between consecutive efforts. The technique used was performed in accordance with the adopted norms [37]. A minimal leak of air (shutter module) was used to prevent blocking of the epiglottis. This minimal leak does not influence the measurements recorded. The inspiratory or expiratory effort was sustained for a minimum of 1 s [38]. 

#### 2.3.3. Aerobic Endurance Measurement

The assessment or measurement of aerobic endurance is achieved by determining the maximum oxygen consumption (VO₂max). Laboratory tests using step-like or ramped protocols with incremental intensities can be used for accurate measurement of VO₂max [39,40]. Because the current experiment could not measure VO₂max directly, the researchers decided to use a cardiorespiratory test, specifically the Cooper test [41,42], and then estimate the achieved results into VO₂max. The formula was as follows: (distance covered in meters−504.9)/44.73 [41]. The Cooper test was conducted in the 400-m track stadium with all players running together, and the total distance covered by each player was recorded. An electronic timer display was placed on the side of the track so the players were familiar with their time. At the end of each lap (every 400 m), the coach told each player his number of laps to run. In addition, markers were set at 100-m intervals around the track to help trainers accurately measure the distance covered at the end of 12 min. To reach VO₂max at 55–57 mL /kg/min (calculated from the run), the player had to get a result over 3 km in the Cooper test. Calculated in terms of time, this gives a pace of 4 min/km, i.e., 80–85% of the intensity for maximum running effort. We are aware that this is an indirect measurement, however, the results achieved by young soccer players are comparable to Reilly’s [43] suggestion that a VO₂max greater than 60 mL/kg/min is required at elite levels of soccer. Before the test, participants conducted their regular 20 min warm-up consisting of a 10-min light run, stretching, running drills, and 100-m rhythm strides. Heart rate was monitored with a Polar RS300X GPS (Polar Electro Oy, Kempele, Finland). The validity of the Cooper test as a correlation between VO_2_max and the distance covered during a 12-min run was 0.94.

#### 2.3.4. Inspiratory Muscle Training

Immediately prior to IMT, instructional classes were held to ensure proper use of the hand-held Threshold IMT device (Philips). Each player received an individual training load according to the results of their preliminary PImax assessment. Due to the nature of the training device, inspiratory muscle strength was converted from kPa to cm H_2_O (1 kPa = 10.2 cm H_2_O). The IMT group performed dynamic inspiratory efforts that increased in frequency during the study period (five repetitions in week 1, 15 repetitions in week 8), twice daily from Monday to Friday for eight weeks (total of 80 sessions) against a pressure threshold load equivalent to 40% MIP in the first week. After the initial setting of the training load at 40% MIP, participants in the IMT group were instructed to periodically increase the load from 45% MIP in week 2 to 75% MIP in week 8 (Table 1). One repetition was the equivalent of a 1-min work period, where the participants exercised for 45 s followed by a 15-s break. The first training session was held at home. The players trained alone each morning between 7:00 a.m. and 9:30 a.m. The second session was executed in the evening (approximately 6:30 p.m.) immediately after the regular football practice. Three times a week during the intervention period, the evening session training was conducted under the supervision of a physiotherapist to ensure a correct technique and appropriate load. The training took place in a sitting position. The players set up an appropriately selected training load, then put on a nose clip and gripped the mouthpiece with their lips. After this, the participants performed fast, energetic inhalations and slow, quiet exhalations. Participants were requested to complete a daily training diary (load) for IMT training throughout the study.

#### 2.3.5. Incremental Endurance Training 

Incremental endurance training is a periodised form of endurance running training that aims to increase the player’s endurance. The IET training protocol included four weeks of endurance running training, consisting of a 6-km steady run and 10–15 × 100 m stride rhythm (65–70% of maximum intensity) executed at the end of training. This training was performed once per week. The initial work rate was based on participants’ known work capacity. The soccer players were asked to start running with a 5.30 min/km pace, ending in week 4 with a 5.0 min/km pace. The work rate increments for each subsequent week increased by 5–10 s/km. Heart rates were monitored using a Polar RS300X GPS HR monitor (Polar Electro Oy, Kempele, Finland). During the endurance run, the players’ HR should not exceed 135–140 bpm, which is a typical manifestation of aerobic performance. All participants ran together, as both groups of the players of both groups (experimental and control) belong to the same team and carry out joint special soccer training for the whole team. After four weeks, participants started the second type of training. This also featured an incremental training load consisting of a 2-km steady run followed by submaximal interval training consisting of three series of five 200 m run repetitions at 75–80% maximum intensity. The work rate increments increased from 45 s per 200 m in week 5 to 38 s per 200 m in week 8 (Table 1). Heart rate was also monitored for each 200 m repetition. They started with a target of 150 bpm and ended with 165 bpm. Interval training for both groups took place at the same time, but control and experimental groups trained separately. Each 200-m repetition was performed in pairs of two players in the same group. Pairs were matched according to the coach’s specifications, which was based on the individual running capabilities of players.

### 2.4. Statistical Analysis

Descriptive statistics (mean ± SD) were calculated for all dependent variables. The Shapiro–Wilk test indicated a normal distribution for all variables. Comparisons between pulmonary function, maximal inspiratory pressure, and Cooper test performance pre- and post-IMT intervention were examined by two-way analysis of variance (ANOVA). Duncan post-hoc tests were performed to determine pairwise differences when significant F ratios were obtained. A comparison of somatic features between the groups was carried out by Student’s *t*-test. The relationship between variables was determined using the Pearson product-moment correlation. The level of significance for all statistical comparisons was set at *p* < 0.05.

## 3. Results

There were no significant differences between groups in terms of their physical characteristics, which confirms their homogeneity (Table 2).

The interaction of the ANOVA was noted for parameters: distance, VO2, PI_max_ (kPa and %), PE_max_ and for FVC (%). For all this parameters *p* = 0,00. Effect by time was noted for parameters distacne (*p* = 0,00), VO2 (*p* = 0,00), FEV_1_ (*p* = 0,00) and FVC (l) *p* = (0,02). Effect by group was noted for parameters PI_max_ (kPa) (*p* = 0,00), PI_max_ (%) (*p* = 0,02). For the other lung function parameters and for HR no significant interactions were noted.

The data for pulmonary function and respiratory muscle strength are shown in Table 3. Significant differences were found between groups for PI_max_ (*p* = 0.0026) and PE_max_ (*p* = 0.0046) after 8 weeks. Additionally, improvements in PI_max_ (62. 8%) and PE_max_ (100%) were observed in the IMT group. There were significant increase changes in FVC and FEV_1_ in the IMT and only significant increase FEV_1_ in control group. Although PEF is a key surrogate marker for expiratory muscle strength, neither the IMT nor control group showed a significant improvement in PEF (*p* = 0.120 and *p* = 0.502, respectively) during the eight-week period.

After IMT and IET training, the experimental group showed a 5.06% increase in running test distance, which translated into a 150.30 m improvement. This improvement was statistically significant (*p* = 0.00). The control group, which only performed the IET training, also showed statistically significant improvement (*p* = 0.00) with an increase of 2.1%, increasing their running distance by 77.50 m (Table 4). An increase in the cardiorespiratory performance of both groups occurred despite the lack of significant differences in HR during a pre-and post-running test performance (Figure 1, Figure 2). The estimated post-run VO_2_max was improved in both groups, with the experimental and control groups showing a 6% (*p* = 0.0001) and 2.5% (*p* = 0.0002) increase, respectively (Table 5). The use of effect size analysis (Cohen’s d) confirmed a significant increase in both measurements, the Cooper test distance and VO_2_max, in the experimental group (d = 2.06 and d = 3.2, respectively) compared to the control group (d = 0.74. and d = 0.63, respectively).

Inter-individual differences in IMT were significantly correlated with a few variables. The relative change in forced expiratory volume in 1 second (FEV_1_) after IMT was related to post-intervention Cooper test performance (r = 0.71). Strong relationships post-intervention were also found in the IMT group between VC and FVC, and FVC and FEV_1_ (r = 0.81 and r = 0.71, respectively).

## 4. Discussion

This study aimed to assess the impact of eight weeks of inspiratory training in addition to a regular program of preseason soccer training, including IET, on changes in pulmonary function, lung ventilation, and aerobic performance in young soccer players. Our results show that the application of an additional IMT program significantly increased the inspiratory and expiratory muscle strength and improved aerobic tolerance, which was directly related to players’ improvement in endurance capacity. In contrast, there was no significant improvement in lung function parameters, with only a small increase in most of them. However, it should be emphasised that soccer players’ basic pulmonary function parameters, measured prior to the intervention, were close to the physiological limits. The starting values of main parameters of lung ventilation were above 100% of the norm level. Moreover, the scope of experiment was designed in order to be possible performed on athletes of all ages.

### 4.1. Effects of Inspiratory Muscle Training on Inspiratory Muscle Function and PI_max_

Application of eight weeks of IMT training resulted in a two-fold increase in inspiratory muscle strength. Implementation of IMT also significantly improved the players’ expiratory muscle strength, as measured by the PE_max_. Differences between the strength of inspiratory and expiratory muscles are common, especially in clinical diagnosis. The extent of these differences is demonstrated in study by Aznar-Lain et al. [44], who observed an increase of 28% in maximal inspiratory pressure (MIP). Moreover the study on rowers showed an increase in MIP of about 33.9% [24]. 

In our experiment, the 100% increase in PE_max_ observed in the experimental group was a surprising finding. This significant improvement could be explained by weak respiratory muscles prior to the application of IMT training. Furthermore, according to the literature expiratory muscles also function in inspiration during forced ventilation [25]. In our study, deep breath also may have changed PE_max_ during IMT.

The results obtained by Lemaitre et al. [45] are similar to ours. These authors employed an 8-week respiratory muscle training program using the SpiroTiger on swimmers specialising in 50- and 200-m sprints. Among the lung function parameters tested, only the forced vital capacity showed a significant increase. Similar results in inspiratory muscle strength and exhalation can be found in the literature [45,46]. Romer et al. [47] studied IMT with varying degrees of resistance for 18 weeks using the POWERbreathe device. Participants were divided into four groups according to the resistance applied: low resistance, medium resistance, high resistance, and control group (no IMT). The groups that underwent IMT training demonstrated an increase in respiratory muscle strength. The greatest increase was observed in the group with medium resistance in the IMT. In contrast, the control group did not show an increase in muscle strength, and some participants even demonstrated a reduction [48]. Interesting results were presented by Volianitis et al. [49], who assessed the value of PI_max_ among 14 rowers before and after 11 weeks of IMT with the POWERbreathe device. In group 1, the resistance accounted for 50% of the inspiratory pressure (PI_max_) and the training consisted of 30 breaths, two times per day. However, in group 2m, there was only 15% resistance. The rowers performed 60 breaths once per day. The inspiratory muscle strength in group 1 increased by 44 ± 25 cm H_2_O (45.3 ± 29.7%), while the PI_max_ in group 2 increased by only 6 ± 11 cm H_2_O (5.3 ± 9.8%). They found a significant improvement in exercise tolerance in both groups [49]. These results indicate that respiratory muscle training is an important complement to daily sports training because it improves the exercise ability of athletes. 

Peak expiratory flow (also known as PEFR) is a key surrogate marker of pulmonary function. In our experiment, neither the IMT nor control group showed a significant improvement in PEF. Because our research was performed on a healthy population who also practiced sport, we could not expect a large improvement in the PEF factor when the output level was already high compared to a healthy population or patients. A reduction in baseline inspiratory and expiratory muscle strength and pulmonary function in patients likely results from neuromuscular disease/restrictive pulmonary syndrome [50]. Therefore, any respiratory muscle training as a resistive exercise program can significantly improve PEF [34,51,52]. Our research did not confirm these results because eight weeks of IMT did not significantly improve the PEF. The improvement was 12.6% (experimental group) and 9% (control group), respectively, for the groups. PEF has also been reported to correlate with height and weight [50]. A good correlation was found between PEF and maximum insufflation capacity (MIC) (r = 0.72) [34]. Because our research was performed on a healthy population who also practiced sport, it was difficult to expect a large improvement in PEF when the output level is already high.

### 4.2. Mechanisms of Improvement in Aerobic Endurance Performance 

The present study showed that 8 weeks of IMT using the Threshold IMT Philips Respironics device could enhance soccer players’ aerobic endurance, as determined by the conversion of running distance from the cardiorespiratory Cooper test into VO₂max. Relative changes in the endurance performance for the IMT group alone (5.06%) are comparable with results of previous respiratory muscle training studies conducted on athletes of different sports. The experimental group had also performed eight weeks of IET training based on running, with the intensity reaching 85% of the HRmax. The control group performed only IET training, and while improvement in the Cooper test was only equal to 2.2%, this represented a significant 3% difference (*p* > 0.05). These small percentage changes were significant, indicating that both groups of players achieved almost maximum results in the Cooper test, approaching the temporary saturation of the players’ endurance capacity. Following the Cooper test, the experimental and control groups achieved a VO_2_max of 58.84 and 55.95 mL·kg1·min^−1^, respectively, which is comparable to other research. Reilly et al. [43] reported that VO_2_max values can range between 55 and 68 mL·kg1·min^−1^, which is primarily due to the inherent differences in tactical positions taken during a match [43] and the fitness level of players [53]. 

Different studies [32,54] show that respiratory muscle training produces different, often contradictory results in athletes who are in training. Mickleborough et al. [54] found that 6 weeks of resistance training of the respiratory muscles among distance runners (roadrunners) led to a significant increase in strength, endurance, maximum power, and the ability to perform inspiratory muscle work [32]. In addition, they found that respiratory muscle training can also change the mechanics of breathing and improve oxygen consumption, ventilation, HR, blood lactate concentration, perceptual response during continuous training load, and the strength of recreational runners [54]. Nicks et al. [55] investigated the effect of respiratory muscle training on respiratory muscle strength, fatigue, and athletic performance in collegiate soccer players. They reported that this type of training had a positive effect on athletic performance. A similar report was presented by Boutellier et al. [56], who found that four weeks of respiratory muscle training improved cycling performance. Romer et al. [47] also evaluated the effects of special inspiratory muscle training on simulated time trial performance in trained cyclists. The results showed the same trend as in previous research, that IMT attenuates the perceptual response to maximal incremental exercise. Consistent with this, Volianitis et al. [49] showed that rowers had better performance in a six-minute rowing test following respiratory muscle training. On the other hand, Pine et al. [57] reported that respiratory muscle training does not affect running sprints. Similar results were found by Morgan et al. [58], who claimed that respiratory muscle endurance training (RMET) improved ventilatory power and endurance, but did not alter VO_2_max or endurance cycling performance among moderately trained male cyclists. Kilding et al. [59] investigated the effect of six weeks of IMT on swimming performance, finding a small positive effect in club-level trained swimmers in events shorter than 400 m. Finally, Sperlich et al. [60] investigated the effect of six weeks of respiratory muscle training on running performance and VO_2_max in a group of German soldiers [60], and found no significant variation in these parameters.

According to Azizimasouleh et al. [61], differences in findings may be due to various factors such as the equipment used for respiratory muscle training, differences in training methods, differences in application of the load (volume and intensity), heterogeneity, and a small number of participants [60]. However, this still does not explain the mechanism of improvement in physical performance. Although IMT has shown to enhance endurance performance, the mechanisms underlying this improvement remain unclear. Nicks et al. [55] indicated that an improvement in PI_max_ does not necessarily explain better performance. Further, while our study showed significant improvements in PI_max_ and PE_max_ in the IMT group, only one strong relationship (r = 0.71) was observed, which was between forced expiratory volume in 1 s (FEV_1_) and post-intervention Cooper test performance. This confirms the assumption of Nicks et al. [55]. They also pointed out that improvement in respiratory muscle function via application of IMT may have an impact on other variables that contribute to performance improvement. However, our study did not show large changes in pulmonary function [55]. In turn, research by Mickleborough et al. [32] on recreational runners found that respiratory muscle training can also change the mechanics of breathing and improve oxygen consumption, ventilation, HR, blood lactate concentration, and perceptual response during continuous training load. In our study, out of all of these factors, we were only able to measure HR in pre- and post-Cooper test. Heart rate measured after completion of the test did not show any differences between groups, despite a greater improvement in the running distance of the IMT group. These findings may reflect an improvement in running economy through endurance running training. Other authors have indicated that as the load (volume and intensity) of exercise increases, the respiratory muscles fatigue substantially [62] and dyspnea also increases. They report that respiratory muscle training reduces dyspnea by restricting the flow of a certain level of minute ventilation. Therefore, respiratory muscle training reduces perceived pressure in the respiratory muscles. Coast et al. [63] indicated that heavy exercise causes respiratory muscle fatigue, which adversely affects athletic performance. This is congruent with our evidence that specific IMT augments the endurance ability of young soccer players. The present study also provides new evidence that enhancements in endurance performance in competitive soccer are possible when IMT is simultaneously carried out with scheduled (periodised) training, focused on the improvement of general and special endurance. It should be noted that the IMT group was selected randomly and accomplished the same level of performance in the pre-intervention Cooper test as the control group. However, only the IMT group achieved a significant increase in running distance in the post-intervention Cooper test. The difference between the groups was 73 m.

Further studies are needed to elucidate the mechanism for the effect of IMT on improved athletics performance, especially endurance performance. A few limitations of our study should be taken into consideration when interpreting the conclusions from our experiment. Firstly, the study should be continued for a longer period and repeated with a larger group. Secondly, future research should further study the influence of respiratory muscle training on intermittent exercise performance, focused either on speed or endurance. Thirdly, more attention should focus on whether respiratory muscle training reduces dyspnea by means of restricting the flow of a certain level of minute ventilation. 

## 5. Conclusions

Eight weeks of IMT had a positive impact on inspiratory (PI_max_) and expiratory muscle strength (PE_max_) in young male soccer players. We also found that increased efficiency of the inspiratory muscles contributes to enhanced improvement in aerobic endurance performance, measured by VO_2_max estimated from the running distance achieved in the cardiorespiratory Cooper test. In conclusion, the improvement in maximal oxygen uptake achieved by a combination of IMT training and IET training at a lower intensity (up to 85% of maximum HR) suggests that high intensity training is not necessary (90–95% of maximum HR), as suggested by current research. This highlights the complex nature of the mechanisms underlying changes in inspiratory muscle function and improvements in performance. Inspiratory muscle training can therefore be considered a worthwhile ergogenic aid for club-level competitive soccer players. 

## Figures and Tables

**Figure 1 ijerph-17-00234-f001:**
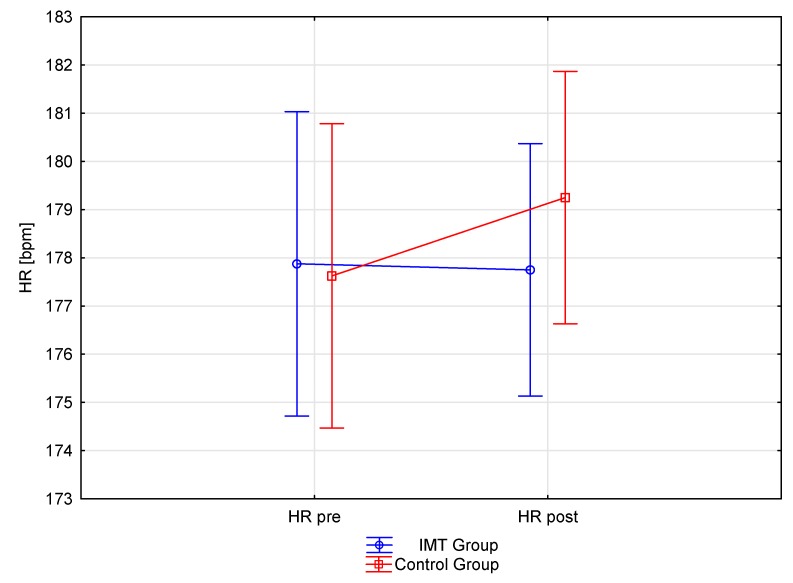
Mean heart rate (HR) measurement pre- and post-inspiratory muscle training (IMT) and endurance training. Vertical bars represent the confidence interval of 0.95.

**Figure 2 ijerph-17-00234-f002:**
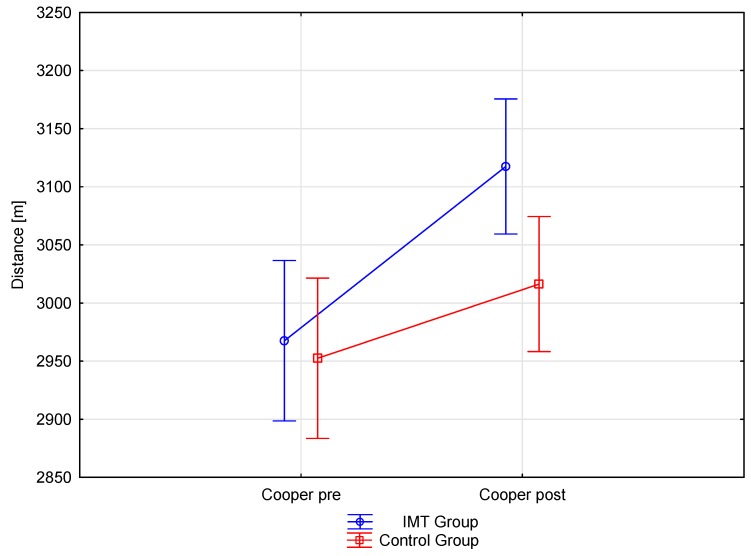
Mean Cooper test measurement pre- and post-inspiratory muscle training (IMT) and endurance training. Vertical bars represent the confidence interval of 0.95.

**Table 1 ijerph-17-00234-t001:** Specification of the load application during eight weeks of inspiratory muscle training (IMT) and incremental running endurance (IRE) workouts.

Training Period	Week 1	Week 2	Week 3	Week 4	Week 5	Week 6	Week 7	Week 8
Inspiratory Muscle Training								
Training load (cmH_2_O)	40% PImax	45% PImax	50% PImax	55% PImax	60% PImax	70% PImax	75% PImax	80% PImax
Time/session (min.)	5	7	9	10	12	13	14	15
Periodized endurance running training								
6 km	5.30 min/km	5.20 min/km	5.10 min/km	5.05 min/km	-	-	-	-
3 × (5 × 200 m)	-	-	-	-	42 s	42 s	40 s	38 s

**Table 2 ijerph-17-00234-t002:** Physical characteristics of young soccer players.

Parameters	IMT Group	Control Group	*p*
	Mean ± SD	Mean ± SD	
Age (year)	17.63 ± 0.48	17.71 ± 0.45	0.7365
Height, cm	182 ± 0.05	180 ± 0.05	0.7881
Mass, kg	68.88 ± 4.48	69 ± 3.34	0.950
BMI	20.89 ± 0.56	21.20 ± 0.84	0.45

**Table 3 ijerph-17-00234-t003:** Statistical characteristics of selected pulmonary function data pre- and post-application of inspiratory muscle training (IMT).

Variables	Experimental Group				Control Group			
	Pre-	Post-	% Differ	d	Pre-	Post-	% Differ	d
VC(l)	5.49 ± 0.56	5.42 ± 0.76	1.18	0.11	4.9 ± 0.83	5.15 ± 1. 05	5.10	0.28
VC (%)	111.13 ± 7.39	107. 08 ± 7.5	3.65	0.56	96.97 ± 15.74	102.14 ± 13.02	5.33	0.38
FVC (l)	5.65 ± 0.75	6.04 ± 0.78	6.90	0.55	5.63 ± 0.69	5.7 ± 0.67	1.24	0.11
FVC (%)	112.98 ± 9.24	121 ± 7.57	7.09	1.01	107.47 ± 13.27	106.13 ± 13.3	1.25	0.10
FEV_1_(l)	4.8 ± 0.57	5.31 ± 0.67	10.62	0.88	4.55 ± 0.81	4.7 ± 0.77	3.29	0.20
FEV_1_ (%)	113.15 ± 10.79	125.11 ± 10.83	10.57	1.18	98.03 ± 12.7	104.71 ± 14.97	1.57	0.51
PEF (l/s)	6.77 ± 1.44	7.7 ± 1.48	13.73	0.73	5.97 ± 1.95	6.36 ± 1.61	6.53	0.23
PEF (%)	82.57 ± 18.31	93 ± 16.9	12.64	0.63	83.56 ± 16.33	91.76 ± 8.08	9.81	0.68
PI_max_ (kPa)	6.85 ± 0.39	11.15 ± 1.07	12.63	5.73	6.39 ± 0.60	7.49 ± 0.86	17.21	1.59
PI_max_ (%)	85.75 ± 5.02	138.08 ± 16.21	61.02	4.66	82.75 ± 3,46	97.08 ± 4,53	17.31	3.80
PE_max_(kPa)	4.02 ± 2.52	8.04 ± 2.26	100	1.79	3.51 ± 1.54	3.93 ± 1.99	3.42	0.25
PE_max_ (%)	84.62 ± 4.7	128.4 ± 14.3	51.73	4.39	83.2 ± 4.5	96.4 ± 3.9	15.86	3.35

Abbreviations: VC—vital capacity, FVC—forced vital capacity, FEV_1_—forced expiratory volume in first second, PEF—peak expiratory flow, PI_max_—maximum inspiratory pressure, PE_max_—maximum expiratory pressure.

**Table 4 ijerph-17-00234-t004:** Comparison of aerobic tolerance assessed by Cooper test, estimated of VO_2_max, and maximum heart rate (HR) performance pre- and post-inspiratory muscle training (IMT). Values are the mean ± SD for all variables.

Variables	IMT-Group				Control Group			
	Pre-	Post-	*p*	d	Pre-	Post-	*p*	d
Distance (m)	2967.50 ± 79.24	3117.50 ± 72.85 *	0.00 *	2.06	2952.50 ± 101.38	3016.25 ± 80.34 *	0.00 *	0.74
VO2max (mL/kg/min)	55.11 ± 1.88	58.41 ± 1.63	0.00 *	3.32	54.75 ± 2.22	55.95 ± 1.80	0.00 *	0.63
HR max (bpm)	177.87 ± 4.73	177.75 ± 3.53	0.87	0.03	177.62 ± 3.50	179.25 ± 3.3	0.07	0.50

Abbreviations: * Significantly higher (*p* < 0.05), VO_2_max - maximal oxygen uptake. Calculated from the running distance.

**Table 5 ijerph-17-00234-t005:** The results of analysis of variance of the parameters and subsequent Duncan’s post hoc test (statistically significant differences are marked in bold).

Variable	IMT Group	Control Group	IMT Group-Control Group	IMT Group-Control Group
	Pre-Post	Pre-Post	Pre-Pre	Post-Post
Distance (m)	**0.0001**	**0.0000**	0.7266	**0.0297**
VO_2_ max (l/min)	**0.0001**	**0.0002**	0.7077	**0.0310**
HR_max_	0.8774	0.0790	0.9037	0.4686
VC (l)	0.7071	0.2179	0.2438	0.5572
FVC (l)	**0.0104**	0.4078	0.9627	0.4080
FVC (%)	**0.0022**	0.1560	0.3172	**0.0321**
FEV_1_ (%)	**0.0457**	**0.0147**	0.3296	0.5938
PEF (1/s)	0.1206	0.5021	0.4158	0.1748
PI_max_ (kPa)	**0.0000**	0.1738	0.7761	**0.0026**
PI_max_ (%)	**0.0001**	0.0848	0.4897	**0.0006**
PE_max_ (kPa)	**0.0001**	0.3616	0.6923	**0.0046**
PE_max_ (%)	**0.0001**	0.4123	0.5976	**0.0049**

Bold -significantly higher (*p* < 0.05).

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
