# Peer review of "The Effect of Respiratory Muscle Training on the Pulmonary Function, Lung Ventilation, and Endurance Performance of Young Soccer Players"

_ijerph, 2019, doi:10.3390/ijerph17010234_

Round 1
Reviewer 1 Report
Dear Authors,
According to me, the article is suitable for publication. A few question marks remain.
Why did you focus on young players precisely? Maybe a sentence or two may be added to make this clear. You should suggest in the conclusion that your experiment (its scope, methods and protocol) was designed in order to bo performed on athletes of all ages and disciplines.Eventually, I know that refs in these kind of articles are a bit messed up, but I raised my eyebrows when I read this passage :
"levels after a cycle of exercises [26]. Research conducted by Archiza 64 et al. [28] on a group of 18 female soccer players indicated that IMT could relieve the metabolism of 65 inspiratory muscles and, as a consequence, improve muscle oxygen supply during high-intensity 66 exercise. This process can be translated into an improvement in fatigue tolerance and running 67 efficiency of soccer players [28]. Consistent with this, Ozmen et al. [27]"
And this sentence is not correct :
"There were significant increase changes in FVC and FEV in the IMT and only significant 232 increase FEV1 in controle group."
Regards,
Reviewer 2 Report
Thank you to the authors, for making many of the suggested changes. This has greatly improved the readability and application of the reported outcomes. I only have few questions remaining:
Section 2.4 (line 216) Why are there two levels of significance? Please indicate which tests use p<0.05 and p<0.01, respectively, with justification for each. Can the authors please reduce the reported p-value to 2 and no more than 3 significant figures after the decimal? Any beyond are not meaningful. If the zeros extend more than 3, it is appropriate to use p<0.001, for example. Results: Please report the interaction of the ANOVA (if any) and If no interaction, then report the main effect by either time or group, which it seems you have reported the main effects but not the interaction. Perhaps interactions did not exist. Line 233: Remove the "e" from the end of "controle". Results: Please provide a reference for how you calculated Cohen's d. Lines 244 and 249: Please change "converted" to "estimated". The VO2max was estimated. Lines 255-258. Please report the level of significance (within 3 decimal points) for any calculated r reported. Table 4 row titles are not in line with the data. Please correct. Table 5 is redundant with information in the text and the ANOVA p-values can be indicated in Table 3 with symbols and foot notes. See other publications for style reference. Line 280-The reference of IMT to influence neural drive seems to be a bit of a stretch especially considering electromyography nor was any other neural outcome measured. Please revise of find sufficient reference to this mechanism. The points considered in the discussion have greatly improved the generalizability in soccer athletes, however as the limitations note, a greater sample would be useful to ensure this is a truly impactful intervention.
Reviewer 3 Report
Yes, the IMT group did show slightly some increase above the control group. However, since there is no significant differences between both groups in VO2max at the post measurement test so it is a negative study.
Reviewer 4 Report
Thank you for your resubmission and addition of highlighted components. Nice job!! I think there is much better structure, content and flow to the article now, however, I still believe this article needs to be read by a native English speaker prior to being published to clean up on grammar errors. There are many phrases and sentences that do not make sense.
For example: "The effectiveness of soccer players depends..."
maximum heart rate (HR) - consider HRmax
"this type of workout is carried out through endurance running performance" - a type of workout is endurance running performance? Consider rewording
"It is based mainly on efforts in aerobic energy expenditure" - consider rewording
"The VO2max" - this is commonly referred to as VO2max and is used to begin sentences alone
"PEF is an individual’s maximum speed of expiration." - should not begin a sentence with acronym/abbreviation
"acknowledged their perfect health to participate in the study." - perfect health - consider revising
"experimental group or control group." - experimental or control group
"maximum oxygen consumption index (VO₂max)." - index?
The Cooper test is a submaximal exercise test done to estimate VO2max
"At the end of each lap (every 400 m), the coach told each 164 player the remaining number of laps to run." - I don't understand - in the Cooper test each individual runs as far as possible in 12 min, correct?
"To reach the converted VO₂max at 55–57 ml/kg/min, soccer players should run over 3 km in the Cooper test." - grammar
"and 10 to 15 100-m rhythm (65–70% of maximum intensity) executed at the end of training." - is there a word missing?
"players are on the same team and perform one joint soccer training." - grammar
Table 1 - "(cmH2O)" - should the "2" be a subscript?
The "p" in p-value is italicized in certain places and not in others - consistency
"This improvement was statistically significant (p = 0.000090)." - not sure there is a need to display p-values to so many decimal places-Authors generally report this as (p<0.01) - same in table 4
I suggest making Figure 1 more easy to read - it is hard to make out what is what - also, the right border is outside
"In contrast, there was no significant improvement in lung function parameters, which showed only an upward trend." - upward trend? Please clarify
"The strong IMT stimuli, applied for a sufficiently long period (8 weeks), led to improvement in the neural drive to the respiratory muscles, resulting in significantly elevated strength." - can you say this if you did not measure neural drive?
"The improvement was 12.6% (experimental group) and 17.21% (control group), respectively, for the groups." - can the authors allude to why the control group may have improved more?
"Relative changes in the endurance performance for IMT group alone (5.06%)" - "for the IMT group alone"?
"achieved almost maximum results 332 in the Cooper test, approaching temporary saturation of the player’s endurance capacity." - I'm not sure I agree that a VO2max of 56-58 ml/kg/min is reaching a saturation for soccer players?
"HR measured after completion..." - I would advise against beginning a sentence with an acronym/abbreviation
Very nice discussion and conclusion.
Author Response
Please see the attachment

This manuscript is a resubmission of an earlier submission. The following is a list of the peer review reports and author responses from that submission.
Round 1
Reviewer 1 Report
Dear authors,
Not being particularly expert in physiology, my comments focus more on the overall relevance of your article in the field of sports science.
The aims of the study only appear at second read and should be emphasised further. Respiratory muscle strength is in my mind intuitively linked to increased respiratory capacity, whereas it appears that your research zeroes in on respiratory stamina. Actually, it makes sense as football requires short, intense dashes repeated over a 90mn interval rather than a long, equally-paced effort such as long-distance running. This should be made clearer in the introduction.
However, as a non-expert, I'm unable to say if your study is original or not. It seems though that the effect of IMT in football lacks documentation.
Methodology is complete and well displayed.
Results and conclusion suffer from the same ills than the introduction: we do not really know if:
you assess the overall usefulness of IMT in football; you try to decipher how IMT can help footballers (under the principle that it is useful no matter what); you want to know how IMT should be performed in football, with which device.It appears that you try to tackle all these issues at once, while at the same time you want to complement what's already been done about IMT in other sports. Fine, but try to entangle all these elements. Specialists of your field may run smoothly through the article but non-experts, even with knowledge in the field, should also be taken into account.
Eventually, there are small mistakes of which I give you a sample:
It is know, that assessing or On the side of the track was placed an electronic timer display the running time The increased of cardio-respiratory I turn a subsequent VO2 max calculation data are shown in Table the control group did not showedEven for a non-native English speaker like me, these mistakes are a nuisance. Besides, your English seems awkward at times, I therefore suggest a moderate editing and revision of the language used in the article.
These minor changes taken into consideration, the article will be according to me ready for publication and certainly a nice read for all those interested in the field of sports physiology.
Regards,
Author Response
Pleas see the attachment

Reviewer 2 Report
This study sought to identify if inspiratory muscle training (IMT) impacted spirometry outcomes or VO2 in youth soccer players. Overall, the manuscript is not written in the best English tense and grammar. The purpose, aims, and hypotheses are not well developed and justification for this research seems to be previously established, thus, not novel. There are several spelling and grammatical errors throughout and several statements that are not clearly written to promote reader comprehension. This is mentioned in several of my comments to the author below.
The use of “soccer” in the title, abstract, keywords, and parts of the manuscript is not consistent with the reference to “football” throughout other components of the manuscript. Please be consistent and identify which sport you are trying to refer to: soccer, futbol, or football. Abstract: please include mean difference values and levels of significance in the abstract. Introduction: The authors discuss the impact of sprinting ability and vaguely mention the goal of training to improve sprint ability. Connecting the dots, I believe what the authors are deducing by this statement it to use IMT to improve sprinting. However, sprinting is anaerobic. The argument in the introduction needs to be better developed to include the notion that high-intensity exercise (at or above ventilatory threshold), repeatedly can enhance VO2 and the translation of that is better oxygen distribution and utilization by the working muscles. Additionally, develop how IMT can potentially enhance these parameters in endurance athletes. Introduction (page 2): The sentence with Gregson (12) is not written in proper grammar. Please revise. Introduction: The use of ‘inspiratory muscle training’ is used several times and is later abbreviated with the words written first. The phrase should be used and the acronym included and subsequently used thereafter. Please review carefully and revise. There are two sentences that are redundant and verbatim in the last two paragraphs of the introduction. Methods: The n-size was outlined in multiple places (participants’ description and experimental design, training section, tables) this is very redundant and unnecessary. Methods: Please provide the company information for Phillips. Methods: Was the experiment blinded? Were other players able to identify who was in a group dissimilar from their own? Methods: Table 2 does not reflect the description provided in section 2.3.5. It is also redundant to have a table and description. Additionally, what is the justification for adding this training for both groups when they were also continuing their typical training? Methods: Although the Cooper test is valid and reliable at predicting VO2max, the test must be individualized. Therefore it is not appropriate for the participants to run the test together because they may pace off each other, use their competitiveness of male teenagers, etc. and the results are rather inaccurate. Methods, Statistical Analysis: What specific type of ANOVA was conducted? Is there a justification for the Dunkan post-hoc tests as opposed, for example, the Bonferroni post-hoc comparisons? The statement about statistical power is confusing. Methods: Can the authors please provide justification for sample size estimation, including calculations.I did not provide further comment due to the complexity of revisions outlined from the Abstract through the Methods. These changes may alter the results and potentially the discussion.
Reviewer 3 Report
PEF is key surrogate marker for expiratory muscle strength.
I don’t understand how PEmax has increased by 100% (4 vs 8 kPa) in IMT group, while there was minimum increase (10%) in PEmax in control group. On the other side, both groups showed similar change in PEF between pre and post measurements.
Considering that both groups showed improvement in distance during Copper test and there was no significant change in VO2max following IMT training. It is clearly that it is a negative study and there is no improve with IMT, the author should rephrase their manuscript accordingly.
Please report % change between pre and post measurements in table.
The control group were supposed to perform the same number of inhalation routine as placebo (no resistance) to better compare between groups. The IMT did extra work and this could account for some differences.
Reviewer 4 Report
Thank you for submitting your article to IJERPH. Here are some comments I hope will strengthen the article.
General comments: Quality of english is poor throughout. Lacks flow and consistency.
Introduction
Is football the same as soccer in this case? In one sentence the anaerobic threshold is 80-90% HR max and in another it's 75% max effort. Be consistent. A lot of background information in the first two paragraphs. Could be shortened considerably. Poor english - poorer oxygen? There are some research? Define BLa You use inspiratory muscle training long before you decide to write (IMT). Then (IMT) is used twice in the introduction. Ozmen ref - says inspiratory capacity increased but this did not translate to tolerance to high intensity exercise. Yet, in the following sentences you say that IMT can increase physical performance? Contradictory? The authors claim "the aim of this study was to assess the impact of eight weeks of IMT on strength improvement and lung ventilation and aerobic endurance performance improvement in young football players." - in the introduction you have already mentioned two studies who have done exactly this? Can it be clarified as to how this is unique? Overall, I don't see enough evidence as to why there is a gap in the literature to support this investigation.Methods
No units - m, cm, kg, lbs What does body height mean? Just height? The soccer training program was recommended...? I don't understand, did somebody take charge and perform this training with the subjects? English language - what does this mean? "In order to reach converted VO₂max about 55-57 ml/kg/min soccer players required run more than 3 km in Cooper test, what gives a pace of 4 min./km (80% or 85% of a maximal effort)." "The heart rates were established to be at a level of 135–140 bips/min, typical for aerobic work."? How can you assign a set target HR for everyone? This is not relative to each individual subject. For example, maybe for certain players 80% of HR max is 150 bpm and for others 70% of HR max is 150 bpm. It seems there are also large spaces between certain words.Results
What is IET??? It suddenly appears in the results section. Graphs need to be clearer. I would suggest labeling HR bpm or beats/min.Discussion
You have a link to a Brianmac in the first paragraph of the discussion. Is this mean to be in the article? Poor overall connection to the introduction and overall premise of the research question.Author Response
Please see the attachment

Round 2
Reviewer 3 Report
Dear Authors, Thanks for your efforts to address my concerns however, I am still not convinced with your revised version. Here is my response to your comments PEF is key surrogate marker for expiratory muscle strength. We did not find information in the literature about the relationship between PEF and PEmax, which is why we did not refer to it in the text and did not want to speculate Please, check this paper and there are many others Expiratory flow maneuvers in patients with neuromuscular diseases https://www.ncbi.nlm.nih.gov/pubmed/16428900 Considering that both groups showed improvement in distance during Copper test and there was no significant change in VO2max following IMT training. It is clearly that it is a negative study and there is no improve with IMT, the author should rephrase their manuscript accordingly. Your statement is imprecise. If the distance in Cooper test has been improved by some section (meters) and it is significant for both groups, which was confirmed by statistical calculations, then if we convert this Cooper test to Vo2max then the results must coincide. So the improvement in the Cooper test for individual groups was the same and was also significant. The differences between the groups occurred in the amount of improvement. Therefore, these are not negative studies. IMT training along with IET running training resulted in greater improvement in Vo2max than just alone IET running training in the control group. The author stating that IET running training resulted in greater improvement in Vo2max (is there significant difference in the post VO2max between the two group?).Reviewer 4 Report
Cardiovascular fitness - why the capital "C"?
It is known, that VO₂max is the most important essential component and is defined as the highest oxygen uptake that can be achieved during dynamic exercise with large muscle groups. - Is this true? VO2max is a better predictor of performance in this population?
Some studies have indicated a significant relation between VO₂max and distance covered during a match. - do you mean relationship?
According to Silva [24], Buttler [25] and, Guy [26], - fix grammar if they are two separate references
which demonstrate that the inclusion of such training in the primary soccer training process can improve maximal inspiratory pressure, exercise tolerance and reduce post exercises blood lactate (BLa) - what does primary soccer training mean? Also post-exercise BLa?
Research conducted by Archiza [28] on a group of 18 female soccer players, indicated that the inspiratory muscle training - why is IMT not abbreviated here?
A problem to be considered when designing a well-balanced Cardiovascular fitness training in soccer - again not sure why "C" is capitlaized?
maximal oxygen uptake development - what does this mean?
great improvement - a large improvement?
It seems reasonable to state that this type of training workout may be represent too strong training stimuli for the junior category of soccer players. - too strong training stimuli? Consider revising.
Therefore, a combination of breathing training with intensity running training, which does not exceed 85 % of anaerobic threshold, should be compensated the same results as previously mentioned (90-95%) intermittent activity. - again grammar - what does intensity running training mean? Also, should be compensated? I don't understand?
Sixteen club-level (Football Academy) competitive junior soccer players (17.63 ± 0.48 years old, 182 ± 0.05 body height, and 68.88 ± 4.48 body mass) participated in this study. - unit of measurement
They were randomly assigned into two groups: inspiratory muscle training (IMT) (n = 8) and control (n = 8). - IMT OR control - Also, you have previously defined IMT so why not just use it? Same for IET?
Medical examinations showed no history of the disease - of the disease? What disease?
The participants were instructed to adhere to their usual diet and not to engage in a strenuous activity the day before testing. - grammar
During the endurance run, the players' heart rate should not exceed 135–140 bits / min, which is a typical manifestation of aerobic performance. - bits/min????
After IMT training and IET training, the experimental group noticed a 5.06% increase in the Cooper’s running distance, - Cooper's running distance
VO2 max (l/min) - VO2max in liters per minute??? I think you mean ml/kg/min please revise
Do the results really need to be drawn out to 6 decimal places? (table 5)
Please fully go back over the entire discussion and paper for editing of the English language.
Double check references - [25] is offset & large space seen with [55].